# Cell Decision Making through the Lens of Bayesian Learning

**DOI:** 10.3390/e25040609

**Published:** 2023-04-03

**Authors:** Arnab Barua, Haralampos Hatzikirou

**Affiliations:** 1Departement de Biochimie, Université de Montréal, Montréal, QC H3T 1C5, Canada; arnab.barua@umontreal.ca; 2Centre Robert-Cedergren en Bio-Informatique et Génomique, Université de Montréal, Montréal, QC H3C 3J7, Canada; 3Center for Information Services and High Performance Computing, Technische Univesität Dresden, 01062 Dresden, Germany; 4Mathematics Department, Khalifa University, Abu Dhabi P.O. Box 127788, United Arab Emirates

**Keywords:** cell decision making, Bayesian learning, least microenvironmental uncertainty principle (LEUP), hierarchical Fokker–Planck equation, cell sensing dynamics, multiscale

## Abstract

Cell decision making refers to the process by which cells gather information from their local microenvironment and regulate their internal states to create appropriate responses. Microenvironmental cell sensing plays a key role in this process. Our hypothesis is that cell decision-making regulation is dictated by Bayesian learning. In this article, we explore the implications of this hypothesis for internal state temporal evolution. By using a timescale separation between internal and external variables on the mesoscopic scale, we derive a hierarchical Fokker–Planck equation for cell-microenvironment dynamics. By combining this with the Bayesian learning hypothesis, we find that changes in microenvironmental entropy dominate the cell state probability distribution. Finally, we use these ideas to understand how cell sensing impacts cell decision making. Notably, our formalism allows us to understand cell state dynamics even without exact biochemical information about cell sensing processes by considering a few key parameters.

## 1. Introduction

Decision making is the process of choosing different actions based on certain goals [1]. Similarly, cells make decisions as a response to microenvironmental signals [2]. When external cues, such as signaling molecules, are received by the cell, a series of chemical reactions is triggered inside the cell [3]. This decision-making process is influenced by intrinsic signal transduction pathways [4], the genetic cell network [5], extrinsic cues [6], and molecular noise [7]. In turn, such intracellular regulation produces an appropriately diverse range of decisions, in the context of differentiation, phenotypic plasticity, proliferation, migration, and apoptosis. Understanding the underlying principles of cellular decision making is essential to comprehend the behavior of complex biological systems.

Cell sensing is a fundamental process that enables cells to respond to their environment and make decisions. Typically, receptors on the cell membrane can detect various stimuli, such as changes in temperature [8], pH [9] or the presence of specific molecules. The specificity of the receptors and the signaling pathways that are activated are critical in determining the response of the cell. However, receptors are not the sole sensing unit of the cell. Recent studies have also revealed that cells use mechanical cues to make decisions about their behavior [10]. For example, cells can sense the stiffness of the substrate they are growing on [11]. In turn, cells make decisions about changing their shape, migration, proliferation or gene expression, in the context of a phenomenon called mechanotransduction [12]. Errors in cell sensing can lead to possible pathologies, such as cancer [13], autoimmunity [14], diabetes [15], etc.

Bayesian inference or updating has been the main toolbox for general-purpose decision making [16]. In the context of cell decision making, this mathematical framework assumes that cells integrate new information and update their internal state based on the likelihood of different outcomes [17]. Although static Bayesian inference was the main tool for understanding cell decisions, recently, Bayesian forecasting has been additionally employed to understand the dynamics of decisions [18]. In particular, Mayer et al. [19] used dynamic Bayesian prediction to model the estimation of the future pathogen distribution by adaptive immune cells. A dynamic Bayesian prediction model was also used for bacterial chemotaxis [20]. Finally, the authors developed the least microenvironmental uncertainty principle (LEUP) that employs Bayesian-based dynamic theory for cell decision making [21,22,23,24].

To understand the stochastic dynamics of the cell-microenvironment system, we focus on the mesoscopic scale and we derive a Fokker–Planck equation. Fokker–Planck formalism was developed to study the time-dependent probability distribution function for the Brownian motion under the influence of a drift force [25]. We can see nowadays a huge number of applications of Fokker–Planck equations (linear and non-linear) across disciplines [26,27]. Here, we will additionally assume a timescale separation between internal and external variables [28]. Timescale separation has been studied rigorously [29] from the microscopic point of view using Langevin equations. In the case of cell decision making, microscopic dynamics have been studied, specifically in the context of active Brownian motion and cell migration using Langevin equations [22,30,31]. Understanding dynamics induced by a timescale separation at the mesoscopic scale, using Fokker–Planck equations, was studied only recently by S. Abe [32].

We will assume a timescale separation, where cell decision time, when internal states evolve, is slower than the characteristic time of the variables that belong to the cellular microenvironment. This assumption is particularly valid for cell decision making at the timescale of a cell cycle, such as differentiation. The underlying molecular regulation underlying these decisions may evolve over many cell cycles [33,34]. When these molecular expressions cross a threshold, the cell decision emerges.

The structure of our paper is as follows: In Section 2 we present the Bayesian learning dynamics for cell decision making. In turn, we derive a fluctuation–dissipation relation and the corresponding continuous-time dynamics of cellular internal states. After that, in Section 3, we elaborate on the concept of the hierarchical Fokker–Planck equation in relation to cellular decision making and the underlying Bayesian learning process. In Section 4, we demonstrate the use of a simple example of coarse-grained dynamics for cell sensing to analyze the steady-state distribution of cellular states in two scenarios: (i) in the absence and (ii) presence of cell sensing. Then, in Section 5, we connect this idea with the least microenvironmental uncertainty principle (LEUP) as a special case of Bayesian learning. Finally, in Section 6, we conclude and discuss our results and findings.

## 2. Cell Decision Making as Bayesian Learning

Cell decisions, here interpreted as changes in the cellular internal states X within a decision time τ, are realized via (i) sensing their microenvironment Y and combining this information with (ii) an existing predisposition about their internal state. In a Bayesian language, the former can be interpreted as the empirical likelihood $P\left(Y|X\right)$ and the latter as the prior distribution $P\left(X\right)$. Interestingly, the previously mentioned distributions are time dependent since we assumed that the cell tries to build increasingly informative priors over time to minimize the cost of energy associated with sampling the cellular microenvironment. For instance, assuming that cell fate decisions follow such Bayesian learning dynamics, during tissue differentiation, we observe the microenvironment evolving into a more organized state (e.g., pattern formation). Therefore, one can observe a reduction in microenvironmental entropy over time, which is further associated with the microenvironmental probability distribution or likelihood in Bayesian inference. Here, we postulate that the cells evolve the distribution of their internal states in the form of Bayesian learning.

### 2.1. A Fluctuation–Dissipation Relation

Formalizing the above, let us assume that after a decision time τ, the cell updates its state from X to $X^{\prime }$ both belonging to $R^{n}$. Moreover, we assume that the microenvironmental variables $Y\in R^{m}$. According to Bayesian learning, the posterior of the previous time $P\left(X|Y\right)$ becomes prior to the next time step, i.e., $P\left(X^{\prime }\right)=P\left(X|Y\right)$. Therefore, the Bayesian learning dynamics read:(1)$$\begin{array}{cc} \; & P\left(X^{\prime }\right)=\frac{P\left(Y|X\right)P\left(X\right)}{P\left(Y\right)},\\ \; & \Rightarrow \ln\frac{P\left(X^{{}^{\prime }}\right)}{P\left(X\right)}=\ln\frac{P\left(Y|X\right)}{P\left(Y\right)}.\\ \; & \Rightarrow \int P\left(X^{{}^{\prime }},X,Y\right)\ln\left(\frac{P\left(X^{{}^{\prime }}\right)}{P\left(X\right)}\right)dX^{{}^{\prime }}dXdY\\ \; & =\int P\left(X^{{}^{\prime }},X,Y\right)\ln\left(\frac{P\left(Y|X\right)}{P\left(Y\right)}\right)dX^{{}^{\prime }}dXdY\\ \; & \Rightarrow {\langle D\left(X^{{}^{\prime }}||X\right)\rangle }_{P(X^{{}^{\prime }}|X)}=\tilde{\beta }I\left(Y,X\right), \end{array}$$
where $\tilde{\beta }=\frac{\int P\left(X^{{}^{\prime }}|X,Y\right)dX^{{}^{\prime }}}{\int P\left(Y|X,X^{{}^{\prime }}\right)dXdY}$, which is different from one if the corresponding conditional distributions require different finite support for their normalization. In the above relation, the Kullback–Leibler divergence $D\left(X^{{}^{\prime }}||X\right)=\int P\left(X^{{}^{\prime }}\right)\ln\left(\frac{P\left(X^{{}^{\prime }}\right)}{P\left(X\right)}\right)dX^{{}^{\prime }}$ that quantifies the convergence to the equilibrium distribution of the internal value of X is connected to the amount of available information $I\left(Y,X\right)=\int P\left(X,Y\right)\ln\left(\frac{P\left(Y|X\right)}{P\left(Y\right)}\right)dXdY$ between the cell and its microenvironment. From Equation (Equation 1), the Kullbeck–Leibler divergence can be further elaborated in terms of Fisher information as
(2)$$\begin{array}{cc} \; & D\left(X^{{}^{\prime }}||X\right)=\int P\left(X^{{}^{\prime }}\right)\ln\left(\frac{P\left(X^{{}^{\prime }}\right)}{P\left(X\right)}\right)dX^{{}^{\prime }}\\ \; & =\int P\left(X^{{}^{\prime }}\right)\ln\left(P\left(X^{{}^{\prime }}\right)\right)dX^{{}^{\prime }}-\int P\left(X^{{}^{\prime }}\right)\ln\left(P\left(X\right)\right)dX^{{}^{\prime }}\\ \; & =\int P\left(X^{{}^{\prime }}\right)\ln\left(P\left(X^{{}^{\prime }}\right)\right)dX^{{}^{\prime }}-\int P\left(X^{{}^{\prime }}\right)\ln\left(P\left(X^{{}^{\prime }}-\Delta X^{{}^{\prime }}\right)\right)dX^{{}^{\prime }}\\ \; & \approx \frac{1}{2}\Delta {X^{{}^{\prime }}}^{T}\int dX^{{}^{\prime }}P\left(X^{{}^{\prime }}\right)\left(\nabla_{X^{{}^{\prime }}}^{2}\ln\left(P\left(X^{{}^{\prime }}\right)\right)\right)\Delta X^{{}^{\prime }}\\ \; & =\frac{1}{2}\Delta {X^{{}^{\prime }}}^{T}F\left(X^{{}^{\prime }}\right)\Delta X^{{}^{\prime }}, \end{array}$$
where $\nabla_{X^{{}^{\prime }}}^{2}$ denotes the corresponding Hessian matrix. Please note that we have assumed very small changes in the internal variable vector X. Here, $F\left(\cdot \right)$ is noted as the Fisher information metric. Since the last formula does not depend on X, then the averaging in Equation (Equation 1) becomes obsolete. Using the relations Equations (Equation 1) and (Equation 2) provides a connection between the Fisher information of the cell internal state and the mutual information with the cellular microenvironment:(3)$$I\left(Y,X\right)=\frac{1}{2\tilde{\beta }}\Delta {X^{{}^{\prime }}}^{T}F\left(X^{{}^{\prime }}\right)\Delta X^{{}^{\prime }}$$

The latter formula implies that the fidelity of the future cell’s internal state is related to the available information in the microenvironment. The above quadratic form makes us view mutual information as a kind of energy functional.

### 2.2. Continuous Time Dynamics

Now, we further assume a very short decision time for the internal variable evolution τ≪1. Along with the Bayesian learning, we assume that the microenvironmental distribution is a quasi-steady state, and therefore we focus only on the dynamics of the internal variable pdf $P\left(X^{\prime })=P(X+\Delta X,t+\tau \right)$, where the increment $\Delta X\in R^{n}$. Using the multivariate Taylor series expansion, we write
(4)$$\begin{array}{cc} \; & P\left(X+\Delta X,t+\tau \right)=\frac{P\left(Y|X,t\right)P\left(X,t\right)}{P\left(Y\right)},\\ \; & \Rightarrow P\left(X,t\right)+\Delta X\cdot \nabla_{X}P\left(X,t\right)+\tau \frac{\partial P\left(X,t\right)}{\partial t}+O\left(\tau^{2},\Delta X^{2}\right)=\frac{P\left(Y|X,t\right)P\left(X,t\right)}{P\left(Y\right)},\\ \; & \Rightarrow \frac{\partial P\left(X,t\right)}{\partial t}\approx -\frac{\Delta X}{\tau }\cdot \nabla_{X}P\left(X,t\right)-\frac{1}{\tau }\left(1-\frac{P\left(Y|X,t\right)}{P\left(Y\right)}\right)P\left(X,t\right) \end{array}$$

The term $\frac{P\left(Y|X,t\right)}{P\left(Y\right)}$ is the information flow due to cell sensing (empirical likelihood). Now, Equation (Equation 4) reaches a steady state only when the cell senses perfectly the microenvironment, i.e., $P\left(Y|X,t\right)$ is equal as $P\left(Y\right)$. The steady solution of the evolution of probability distribution helps us to understand how it evolved over a long time, which can tell us how the internal variables of cells settle. So, close to the steady state (i.e., $\frac{\partial P\left(X,t\right)}{\partial t}=0$), the Equation (Equation 4) further reads as
(5)$$\begin{array}{cc} \; & \Delta X\cdot \nabla_{X}P\left(X\right)=-\left(1-\frac{P\left(Y|X\right)}{P\left(Y\right)}\right)P\left(X,t\right)\approx i\left(X:Y\right)P\left(X\right)\\ \; & \Rightarrow \sum_{i=1}^{n}\Delta X_{i}\frac{\partial }{\partial X_{i}}P\left(X\right)=i\left(X:Y\right)P\left(X\right)\\ \; & \Rightarrow \sum_{i=1}^{n}\Delta X_{i}\frac{\partial }{\partial X_{i}}\left(\ln P\left(X\right)\right)=i\left(X:Y\right) \end{array}$$

Above, we used the identity ln(x)≈1−x for small *x* and the definition of the *point-wise mutual information* as $i\left(X:Y\right)=\ln\frac{P\left(Y|X\right)}{P\left(Y\right)}$.

Deriving an analytical solution for Equation (Equation 5) is a daunting task. Therefore, we use a *Gibbs ansatz*, which additionally assumes that mutual independence of the random variable $X_{i}⊥X_{j}$ for i≠j:(6)$$\begin{array}{cc} \; & P\left(X\right)\approx \prod_{i=1}^{n}P\left(X_{i}\right)=\frac{e^{-\sum_{i=1}^{n}\alpha_{i}U_{i}}}{Z}\\ \; & \Rightarrow P\left(X_{i}\right)=\frac{e^{-\alpha_{i}U_{i}}}{Z_{i}} \end{array}$$

Combining the above ansatz with the Equation (Equation 5), we obtain
(7)$$i\left(X:Y\right)=-\sum_{i=1}^{n}\Delta X_{i}\alpha_{i}\frac{\partial U_{i}\left(X_{i}\right)}{\partial X_{i}}$$

Using our results in the Appendix A and in particular Equation (Equation 38), we can write
(8)$$i\left(X:Y\right)=\ln P\left(Y|X\right)-\ln P\left(Y\right)=\sum_{i=1}^{n}\ln P\left(Y|X_{i}\right)-n\ln P\left(Y\right)=\sum_{i=1}^{n}i\left(X_{i}:Y\right)$$

Now combining the above equations and integrating for the variable $X_{i}$, we can obtain an explicit formula for the potential $U_{i}$:(9)$$U_{i}\left(X_{i}\right)=-\frac{1}{\alpha_{i}\Delta X_{i}}\int^{X_{i}}i\left({\tilde{X}}_{i}:Y\right)\;d{\tilde{X}}_{i}.$$

Therefore, the probability distribution for the internal variable $X_{i}$ reads
(10)$$P\left(X_{i}\right)=\frac{e^{\beta_{i}\int^{X_{i}}i\left({\tilde{X}}_{i}:Y\right)\;d{\tilde{X}}_{i}}}{Z_{i}},$$
where we introduce the *sensitivity* parameter $\beta_{i}\propto \Delta X_{i}^{-1}$. Working out further the above equation, we obtain
(11)$$\begin{array}{cc} \; & P\left(X_{i}\right)=\frac{e^{\beta_{i}\int^{X_{i}}i\left(Y:\tilde{X_{i}}\right)d{\tilde{X}}_{i}}}{Z_{i}}\\ \; & =\frac{e^{\beta_{i}\int^{X_{i}}d{\tilde{X}}_{i}i\left(Y:\tilde{X_{i}}\right)\int_{R^{m}}P(Y|{\tilde{X}}_{i})dY}}{\int dX_{i}e^{\beta_{i}\int^{X_{i}}d{\tilde{X}}_{i}i\left(Y:\tilde{X_{i}}\right)\int_{R^{m}}P(Y|{\tilde{X}}_{i})dY}}\\ \; & =\frac{e^{-\beta_{i}\int^{X_{i}}S\left(Y|X={\tilde{X}}_{i}\right)d{\tilde{X}}_{i}-\beta_{i}\int^{X_{i}}d{\tilde{X}}_{i}\int_{R^{m}}dYP\left(Y|{\tilde{X}}_{i}\right)\ln p\left(Y\right)}}{\int dX_{i}e^{-\beta_{i}\int^{X_{i}}S\left(Y|X={\tilde{X}}_{i}\right)d{\tilde{X}}_{i}-\beta_{i}\int^{X_{i}}d{\tilde{X}}_{i}\int_{R^{m}}dYP\left(Y|{\tilde{X}}_{i}\right)\ln p\left(Y\right)}}\\ \; & =\frac{e^{-\beta_{i}\int^{X_{i}}d{\tilde{X}}_{i}S\left(Y|X={\tilde{X}}_{i}\right)-\beta_{i}^{\prime }X_{i}}}{Z_{i}}. \end{array}$$
where we used the fact that the $\int_{R^{m}}P(Y|{\tilde{X}}_{i})dY=1$ and the definition of the conditional entropy $S\left(Y|X={\tilde{X}}_{i}\right)=-\int dYP\left(Y|X={\tilde{X}}_{i}\right)\ln P\left(Y|X={\tilde{X}}_{i}\right)$. The parameter $\beta_{i}^{\prime }=\beta_{i}d{\tilde{X}}_{i}\int_{R^{m}}dYP\left(Y|{\tilde{X}}_{i}\right)\ln p\left(Y\right)$ is a real constant.

## 3. Connection between Hierarchical Fokker–Planck Equation and Bayesian Learning Process

In this section, we shall discuss the connection between dissipative dynamics and Bayesian learning regarding the cell decision-making process. Since cell decision making is a stochastic process of the continuous internal variable X, we can assume the existence of the Fokker–Planck description. When there exists a timescale separation between two dynamical variables, a hierarchical Fokker–Planck equation [32] can be derived. In this section, we shall show how this formalism can be applied in cell decision making and also will show how it helps us to study the origin of biophysical forces in terms of the information-theoretic quantities as shown in Figure 1.

Let us consider X and Y to be the internal variables which evolve in a slow timescale and external variables that are fast, and the corresponding 2-tuple random variables (which evolve over time) as
(12)$$M=\left(\begin{array}{c} M_{1}\\ M_{2} \end{array}\right)=\left(\begin{array}{c} X\\ Y \end{array}\right)$$

Now for a random variable M, one can write in the Ito-sense the generalized stochastic differential equation for multiplicative noise processes as
(13)$$dM=K\left(M,t\right)dt+\Sigma \left(M,t\right)dW$$

In this above Equation (Equation 13), we define the drift term K, the Σ that is a 2×2 covariance matrix and dW as the Wiener process [35], which satisfies the mutual independence condition below
(14)$$dW_{i}dW_{j}=\delta_{ij}dt$$

The realization of $X=M_{1}$ and $Y=M_{2}$, obeys the time-dependent joint probability. $P\left(X,Y,t\right)$ which satisfies the generalized Fokker–Planck equation. Now, the generalized Fokker–Planck equation [35,36,37] corresponding to the Langevin Equation (Equation 13) for two-variable homogeneous processes can be written as
(15)$$\frac{\partial P}{\partial t}=-\sum_{p=1}^{2}\frac{\partial }{\partial M_{p}}\left(K_{p}P\right)+\sum_{p,q=1}^{2}\frac{\partial^{2}}{\partial M_{p}\partial M_{q}}\left(\sigma_{pq}P\right)$$
where drift coefficients $K_{p}=K_{p}\left(X,Y,t\right)$ and diffusion coefficients $\sigma_{pq}=\sigma_{qp}=\sigma_{pq}\left(X,Y,t\right)$.

The Fokker–Planck equations represent the mesoscopic scale of a dynamical system [38]. Interestingly, in a large timescale separation at the mesoscopic level, the degrees of freedom associated with the fast variables depend on slow variables but not vice versa. Since we assumed that the microenvironmental variables Y evolve at the fastest timescale, it follows that $K_{1}\equiv K_{1}\left(X,Y\right)$, $K_{2}\equiv K_{2}\left(X\right)$ and $\sigma_{22}\left(Y,X,t\right)\equiv \sigma_{22}\left(X\right)$. To use the separation method adiabatically, we shall substitute
(16)$$P\left(X,Y,t\right)=P\left(Y,t|X\right)P\left(X\right),$$
where the P(X) is time invariant relative to the evolution of the microenvironmental variables. Thus, the dynamics of the joint probability reduces to the dynamics of the fast variable Y and using Equation (Equation 15), we have
(17)$$\begin{array}{cc} \; & \frac{\partial P\left(X,Y,t\right)}{\partial t}=P\left(X\right)\frac{\partial P\left(Y,t|X\right)}{\partial t}=-P\left(X\right)\nabla_{Y}\cdot \left(K_{1}\left(Y,X,t\right)P\left(Y,t|X\right)\right)\\ \; & -\nabla_{X}\cdot \left(K_{2}\left(X\right)P\left(Y,t|X\right)P\left(X\right)\right)\\ \; & +P\left(X\right)\nabla_{Y}^{2}\left(\sigma_{11}\left(Y,X,t\right)P\left(Y,t|X\right)\right)\\ \; & +2\nabla_{X}\cdot \left[P\left(X\right)\nabla_{Y}\left(\sigma_{12}\left(Y,X,t\right)\right)P\left(Y,t|X\right)\right]\\ \; & +\nabla_{X}^{2}\left(\sigma_{22}\left(X\right)P\left(X\right)P\left(Y,t|X\right)\right) \end{array}$$

From this point, the equations for the fast degree of freedom and the others (slow degree of freedom and coupling between them) are derived, respectively, as follows:(18)$$\frac{\partial P\left(Y,t|X\right)}{\partial t}=-\nabla_{Y}\cdot \left(K_{1}\left(Y,X,t\right)P\left(Y,t|X\right)\right)+\nabla_{Y}^{2}\left(\sigma_{11}\left(Y,X,t\right)P\left(Y,t|X\right)\right),$$
(19)$$\begin{array}{cc} \; & \nabla_{X}\cdot \left(K_{2}\left(X\right)P\left(Y,t|X\right)P\left(X\right)\right)+2\nabla_{X}\cdot \left[P\left(X\right)\nabla_{Y}\cdot \left(\sigma_{12}\left(Y,X,t\right)\right)P\left(Y,t|X\right)\right]\\ \; & +\nabla_{X}^{2}\left(\sigma_{22}\left(X\right)P\left(X\right)P\left(Y,t|X\right)\right)=0 \end{array}$$

From Equation (Equation 19), if we integrate once over X, it follows
(20)$$\begin{array}{cc} \; & -K_{2}\left(X\right)P\left(Y,t|X\right)P\left(X\right)+2P\left(X\right)\nabla_{Y}\cdot \left(\sigma_{12}\left(Y,X,t\right)P\left(Y,t|X\right)\right)\\ \; & +\nabla_{X}\cdot \left(\sigma_{22}\left(X\right)P\left(Y,t|X\right)P\left(X\right)\right)=0, \end{array}$$
and working further on the equations
(21)$$\begin{array}{cc} \; & -K_{2}\left(X\right)P\left(Y,t|X\right)P\left(X\right)\\ \; & +2P\left(X\right)\nabla_{Y}\cdot \left(\sigma_{12}\left(Y,X,t\right)\right)P\left(Y,t|X\right)+2P\left(X\right)\sigma_{12}\left(Y,X,t\right)\nabla_{Y}\left(P\left(Y,t|X\right)\right)\\ \; & +\nabla_{X}\cdot \left(\sigma_{22}\left(X\right)\right)P\left(Y,t|X\right)P\left(X\right)+\sigma_{22}\left(X\right)P\left(X\right)\nabla_{X}\left(P\left(Y,t|X\right)\right)\\ \; & +\sigma_{22}\left(X\right)P\left(Y,t|X\right)\nabla_{X}\left(P\left(X\right)\right)=0. \end{array}$$

To isolate the slow degree of freedom, we further separate Equation (Equation 21) as follows:(22)$$-\left(K_{2}\left(X\right)-\nabla_{X}\cdot \left(\sigma_{22}\left(X\right)\right)\right)P\left(X\right)+\sigma_{22}\left(X\right)\nabla_{X}\left(P\left(X\right)\right)=0,$$
(23)$$2\nabla_{Y}\cdot \left(\sigma_{12}\left(Y,X,t\right)P\left(Y,t|X\right)\right)+\sigma_{22}\left(X\right)\nabla_{Y}\left(P\left(Y,t|X\right)\right)=0,$$
which are the equations for the slow degree of freedom and the coupling, respectively. Thus, Equations (Equation 18), (Equation 22) and (Equation 23) are the ones to be analyzed. Now, we try to establish the connection between hierarchical Fokker–Planck equations and steady-state Bayesian learning when the internal variable is one-dimensional. The general solution of Equation (Equation 22) in one dimension can be written as
(24)$$P\left(X_{i}\right)=f_{0}\exp\left(\int^{X_{i}}d{\tilde{X}}_{i}\frac{K_{2}\left({\tilde{X}}_{i}\right)}{\sigma_{22}\left({\tilde{X}}_{i}\right)}-\ln\sigma_{22}\left(X_{i}\right)\right).$$
where $f_{0}$ is a positive constant. If we have information about the drift term $K_{2}\left({\tilde{X}}_{i}\right)$ and diffusion coefficient $\sigma_{22}\left(X_{i}\right)$, we can easily calculate the probability distribution of the internal variables from Equation (Equation 24), which is independent of the fast variable. So, comparing Equations (Equation 11) and (Equation 24), one can obtain
(25)$$\begin{array}{cc} \; & P\left(X=X_{i}\right)=\frac{e^{-\beta_{i}\int^{X_{i}}d{\tilde{X}}_{i}S\left(Y|X={\tilde{X}}_{i}\right)-\beta_{i}^{\prime }X_{i}}}{Z_{i}}=f_{0}\exp\left(\int^{X_{i}}d{\tilde{X}}_{i}\frac{K_{2}\left({\tilde{X}}_{i}\right)}{\sigma_{22}\left({\tilde{X}}_{i}\right)}-\ln\sigma_{22}\left(X_{i}\right)\right),\\ \; & \Rightarrow \frac{e^{-\beta_{i}\int^{X_{i}}d{\tilde{X}}_{i}S\left(Y|X={\tilde{X}}_{i}\right)-\beta_{i}^{\prime }X_{i}}}{Z_{i}}=\tilde{f}\exp\left(\frac{1}{\sigma_{22}}\int^{X_{i}}K_{2}\left(\tilde{X_{i}}\right)d\tilde{X_{i}}\right),\\ \; & \Rightarrow -\beta_{i}\int^{X_{i}}d{\tilde{X}}_{i}S\left(Y|X={\tilde{X}}_{i}\right)-\beta_{i}^{\prime }X_{i}=\ln\left[\tilde{f}Z\right]+\frac{1}{\sigma_{22}}\int^{X_{i}}K_{2}\left(\tilde{X_{i}}\right)d\tilde{X_{i}},\\ \; & \Rightarrow K_{2}\left(X_{i}\right)=-\beta_{i}\sigma_{22}S\left(Y|X=X_{i}\right)-\frac{\beta_{i}^{\prime }}{2}X_{i}^{2}. \end{array}$$

In the above Equation (Equation 25), $\tilde{f}$ is defined as $\frac{f_{0}}{\sigma_{22}}$ and the diffusion coefficient $\sigma_{22}\left(X_{i}\right)$ in Equation (Equation 25) is considered constant, i.e., $\sigma_{22}\left(X_{i}\right)=\sigma_{22}$. Therefore, we can directly see how the microenvironmental entropy and the drift force have a unique relation.

## 4. Implications of Cell Sensing Activity

Cell sensing is usually defined as a process where cells communicate with the external environment based on their internal regulatory network of signaling molecules. In the context of Bayesian learning cells, the cell sensing distribution P(Y|X) plays a central role. The problem is that the regulation between a particular sensing molecule and the set of microenvironmental variables can be complex [39]. For simplicity, we constrain ourselves to one-dimensional internal and external variables. Let us consider that the microenvironment *Y* is sensed by the internal state *X* as
(26)$$Y_{X}=Y|X=F\left(X,\langle Y^{n}\rangle \right).$$

Here, we assume that the cell sensing function F(·) also depends on moments of the microenvironmental variable and, consequently, we assume their existence. Now, if we perform a Taylor series expansion around the mean value of the internal state $\bar{X}$ in Equation (Equation 26),
(27)$$\begin{array}{cc} \; & Y_{X}=F\left(\bar{X}\right)+|\frac{\partial }{\partial X}F\left(\bar{X}\right)|\left(X-\bar{X}\right)\\ \; & Y_{X}-\bar{Y}=F\left(\bar{X}\right)-\bar{Y}+|\frac{\partial }{\partial X}F\left(\bar{X}\right)|\left(X-\bar{X}\right)\\ \; & \sigma_{Y|X}^{2}\left(x\right)={\langle {\left(b+g(x-\bar{X})\right)}^{2}\rangle }_{P(Y)}={\left(b+g(x-\bar{X})\right)}^{2} \end{array}$$

Here, we define the bias term $b=F\left(\bar{X}\right)-\bar{Y}$ and the linear sensing response to microenvironmental changes *Y* defined by $g=|\frac{\partial }{\partial X}F\left(\bar{X}\right)|$. Please note that both *b* and *g* depend only on the moments of *Y*. The biological relevance of this linear sensing function can be found in the classical receptor–ligand models [40]. In particular, let us assume that the sensed environment variable Y|X is the ligand–receptor complex and the variable *X* corresponds to the receptor density. If *g* is a first-order Hill function for the first moment of *Y*, which in this context is the ligand concentration, and if $F(\bar{X})=0$, then first, Equation (Equation 26) corresponds to the textbook steady state of the complex formation [40].

Moreover, we consider the microenvironmental distribution as Gaussian, where the entropy of the microenvironment, conditioned by the corresponding internal states, can be written as
(28)$$S\left(Y|X=x\right)=\frac{1}{2}\ln\left(2\pi e\sigma_{Y|X}^{2}\left(x\right)\right).$$

Now, using the above expression of microenvironmental conditional entropy, one can calculate the steady state of cellular internal variables from Bayesian learning using Equation (Equation 11). In turn, it can be written as
(29)$$\begin{array}{cc} \; & P\left(X\right)\propto e^{-\beta \int^{X}S\left(Y|X=\tilde{X}\right)d\tilde{X}-\beta^{{}^{\prime }}X}\\ \; & =e^{-\beta \int^{X}\ln\left(b+g(\tilde{X}-\bar{X})\right)d\tilde{X}-\beta^{{}^{\prime }}X} \end{array}$$

Interestingly, we have two cases to study the steady-state distribution of the cellular internal states: (I) when the response of *X* to microenvironmental changes is negligible and (II) when there exists a finite correlation value between internal cellular state and microenvironmental state, which follows as
(30)$$\begin{array}{cc} \; & P\left(X\right)=C_{1}e^{-\bar{\beta }X},\;g\ll 1\\ \; & P\left(X\right)=C_{2}{\left(b+g\left(X-\bar{X}\right)\right)}^{\beta \left(X-\bar{X}+\frac{b}{g}\right)}e^{-(\beta +\beta^{{}^{\prime }})X},\;g=O\left(1\right) \end{array}$$

Here, $C_{0}$ and $C_{1}$ are normalization constants of corresponding probability distributions, and $\bar{\beta }$ is defined as $\left(\beta \ln b+\beta^{{}^{\prime }}\right)$. In case (I), i.e., when *g* is equal to 0, the steady-state distribution of internal variables converges to an *exponential* distribution. Please note that the sensor OFF probability distribution makes sense only for $\bar{\beta }>0$. In the ON case, when the linear response *g* is finite and β<0, the expression of the steady state is *unimodal*. Interestingly, for β>0 and for a finite range of *X* values, the distribution is *bimodal* with the highest probability density around the boundaries of the domain. Please note that for very large $\beta^{{}^{\prime }}$ values, the exponential decay term dominates. In a nutshell, the above expression of the internal state shows how an ON-OFF switching case can happen when the environment correlates with the cell and as a response the cell senses the microenvironment changing its phenotype, which confirms the existence of the monostable–bistable regime as shown in Figure 2.

## 5. Bayesian Learning Minimizes the Microenvironmental Entropy in Time

Recently, we postulated the least environmental uncertainty principle (LEUP) for the decision making of cells in their multicellular context [21,22]. The main premise of LEUP is that the *microenvironmental entropy/uncertainty decreases over time*. Here, we hypothesized that cells use Bayesian learning to infer their internal states from microenvironmental information. In particular, we previously showed that $\frac{dS(Y|X)}{dt}\leq 0$ [22], which is the case in the Bayesian learning case. To illustrate this, let us focus on the Gaussian 1D case of the previous section. Averaging Equation (Equation 27) for the distribution p(X,Y), we can obtain the following:(31)$$\sigma_{Y|X}^{2}=b^{2}+g^{2}\sigma_{x}^{2}.$$

One can show that the linear response term is proportional to the covariance of the internal and external variables, i.e., g∝cov(X,Y) as a result of the Gaussian conditional variable. As the Bayesian learning is reaching equilibrium, according to Equation (Equation 1), the covariance approaches zero and consequently, $\sigma_{Y|X}^{2}\overset{t\to \infty }{\to }b^{2}.$

Please note that we still assume that the microenvironmental pdf is in a quasi-steady state due to the time scale separation [22]. The latter implies that the variance of Y|X is monotonically decreasing and therefore S(Y|X) is also a decaying function in time. Therefore, we can postulate that Bayesian learning is compatible with the LEUP idea.

Mathematically speaking, the original LEUP formulation was employing an entropy maximization principle, where one can calculate the distribution of cell internal states using as a constraint the mutual information between local microenvironment variables and internal variables. Adding as a constraint the expected value of internal states, the corresponding variational formulation reads:(32)$$\begin{array}{cc} \frac{\delta }{\delta P\left(X_{i}\right)} & \{S\left(X_{i}\right)+\beta_{i}\left[\int dX_{i}P\left(X_{i}\right)\int dYP\left(Y|X_{i}\right)i\left(Y:X_{i}\right)-\bar{I}\left(Y:X_{i}\right)\right]\\ \; & -\beta_{i}^{{}^{\prime }}\left[\int P\left(X_{i}\right)X_{i}dX_{i}-\bar{X_{i}}\right]-\lambda_{i}\left[\int P\left(X_{i}\right)dX_{i}-1\right]\}=0, \end{array}$$

Here, $\delta /\delta P\left(X_{i}\right)$ is the functional derivative with respect to the internal states. Three Lagrange multipliers in Equation (Equation 32), i.e., $\beta_{i}$, $\beta_{i}^{{}^{\prime }}$ and $\lambda_{i}$, are associated with the steady-state value of the mutual information $\bar{I}\left(Y,X_{i}\right)$, mean value of the internal variables and the normalization constant of the probability distribution. The constraint or the partial information about the internal and external variables is written in terms of the statistical observable. Solving Equation (Equation 32), we can find a *Gibbs*-like probability distribution:(33)$$P\left(X_{i}\right)=\frac{e^{\beta_{i}\;D\left(Y|X={\tilde{X}}_{i}\left|\right|Y\right)-\beta_{i}^{{}^{\prime }}X_{i}}}{Z_{i}}=\frac{e^{-\beta_{i}S\left(Y|X_{i}={\tilde{X}}_{i}\right)-\beta_{i}^{\prime }X_{i}}}{Z_{i}^{{}^{\prime }}}.$$

Here, $Z_{i}^{{}^{\prime }}=\int e^{-\beta_{i}S\left(Y|X_{i}={\tilde{X}}_{i}\right)-\beta_{i}^{{}^{\prime }}{\tilde{X}}_{i}}d{\tilde{X}}_{i}$ is the normalization constants. Please note that we used the fact that $D\left(Y|X=X_{i}\left|\right|Y\right)=-S\left(Y|X=X_{i}\right))-\int dYp\left(Y|X\right)\ln p\left(Y\right)$, where the second term gets simplified since it is independent of $X_{i}$. Interestingly, it can coincide with the Bayesian learning context as a special case, where the $i\left(Y:X_{i}\right)\to 0$. Using Equation (Equation 11) in a finite domain $X_{i}\in \Omega$ and the mean value theorem for integration, there exists a value ${\hat{X}}_{i}$ such that
(34)$$P\left(X_{i}\right)=\frac{e^{-\beta_{i}S\left(Y|X_{i}={\hat{X}}_{i}\right)-\beta_{i}^{\prime }X_{i}}}{Z_{i}^{\prime }}.$$

Therefore, the form of the maximum entropy distribution (Equation 33) and the Bayesian learning steady-state distribution (Equation 11) coincide when the random variable $X_{i}$ takes values in the vicinity of ${\hat{X}}_{i}$.

## 6. Discussion

In this paper, we elaborate on the idea of cellular decision-making based on Bayesian learning, assuming a time-scale separation between environmental and internal variables. We derive a stochastic description of the temporal evolution of the corresponding dynamics, studying the impact of cell sensing on the internal state distribution and the corresponding microenvironmental entropy evolution.

An interesting finding is the steady-state distributions of the internal state depending on the state of the cell sensor activity. When the cell weakly senses its microenvironment, the internal state follows an exponential distribution (see Equation (Equation 29)). In terms of the receptor–ligand sensing mechanism, this implies that no specific amount of receptors is expressed by the cell. When the sensor is in the ON state, then a unimodal distribution occurs, which implies that the cell expresses a precise number of, for example, receptors as a response to a certain stimulus. The former can be viewed as the physiological modus operandi of the cell. However, when the sensitivity β changes sign, then the probability mass is distributed to the extreme values of the internal state space. This can be potentially mediated by a bistability regulation mechanism, e.g., for the receptor production. Such bimodality is relevant in the context of cancer, where it is considered a malignancy prognostic biomarker [41,42]. However, it can also occur in physiological cases such as in healthy immune cells [43]. It would be interesting to explore if the sensing activity is a plausible mechanism for explaining transitions from unimodality to bimodality.

One important point of interest is the range of validity of regarding the timescale separation between the cell decision and the cell’s microenvironmental variables. In particular, we assumed that the internal state characteristic time is slower than the microenvironmental one, which can be true for decision timescales related to the cell cycle duration. Sometimes, cell decisions may seem to be happening within one cell cycle, but the underlying molecular expressions may evolve even over many cell cycles [33,34]. During the cell cycle time, we can safely assume that external variables, such as chemical signal concentrations or migrating cells, will be in a quasi-equilibrium state. However, for cell decisions with shorter timescales, such as migration-related processes, which are at the order of one hour, this assumption needs to be relaxed. In the latter case, the discrete-time dynamics presented in Section 2 are still valid.

Here, we assumed that the fast timescale environmental variables can be influenced by the current state of cellular internal variables. However, we did not consider the influence of the past time states. This would imply non-Markov dynamics for internal cellular state evolution. It would be interesting to study how this assumption could impact the information flow dynamics between environmental states and cellular internal variables.

The outlined theory is related to single-cell decision making. Our ultimate goal is to understand how Bayesian learning impacts the collective behavior of a multicellular system. An agent-based model driven by Bayesian learning dynamics could be used to analyze the collective dynamics as in [22]. Interestingly, we expect a Bayesian learning multicellular theory to produce similar results to the rattling interactions introduced in [44]. Similarly, in rattling dynamics, an approximation of the mutual information between neighboring individuals is minimized, leading to the emergence of a self-organized active collective state.

Regarding cell sensing, we took an agnostic approach, where a generic function was assumed. Linearizing the sensing function leads to steady-state dynamics, which could be seen in the ligand–receptor dynamics, e.g., [45], by assuming our sensed environment variable Y|X is the ligand–receptor complex and the variable *X* the receptors. It will be alluring to further investigate the non-linear relationship between internal and external variables, which means considering a few more terms in the Taylor series expansion of conditional variance to simulate a greater variety of biological sensing scenarios.

Our decision-making approach is a dynamic theory based on Bayesian learning of cellular internal states upon variations of the microenvironment distribution. The classical Bayesian decision-making methods are of a static nature relying on Bayesian inference tools [16]. Belief updating networks resemble the ideas of Bayesian learning; however, such algorithms are treated typically computationally, and to our knowledge there have been not many attempts of deriving dynamic equations [46]. The oldest life science field where such ideas have been developed is human cognition. This dates back to 1860 when Hermann Helmholtz postulated the Bayesian brain hypothesis, where the nervous system organizes sensory data into an internal model of the outside world [47]. Recently, Karl Friston and collaborators formulated the brain free energy theory deriving a variational Bayesian framework for predicting cognitive dynamics. Friston’s ideas were recently translated into the Bayesian mechanics approach [48]. The latter resembles our approach, but it requires concepts of Markov blankets and control theory. The main difference is that all of the above attempt to model human cognition and not cell decision making.

Finally, assuming Bayesian learning/LEUP as a principle of cell decision making, we can bypass the need for a detailed understanding of the underlying biophysical processes. Here, we showed that even by using an unknown cell sensing function, we can infer the state of the cell with a minimal number of parameters. Building on these concepts, we can create theories and predictive tools that do not require the comprehensive knowledge of the underlying regulatory mechanisms.

## Figures and Tables

**Figure 1 entropy-25-00609-f001:**
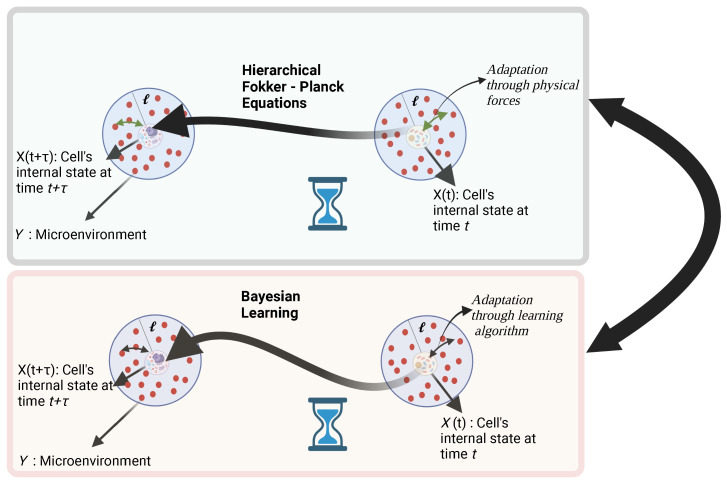
A schematic picture of cellular decision making in a complex microenvironment through physical forces and through Bayesian learning.

**Figure 2 entropy-25-00609-f002:**
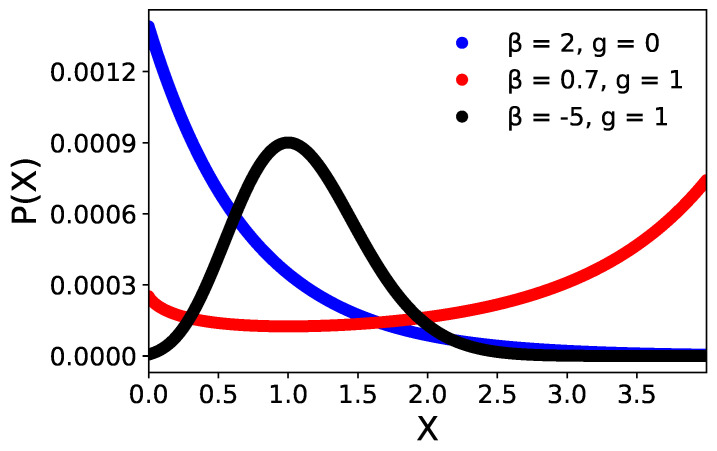
Plot of the normalized steady−state probability distribution of cellular phenotypes for both cases (I)g=0 and (II)g=1 with different values of β. *b* and $\bar{X}$ parameter is kept at 2 and $\beta^{{}^{\prime }}$ is kept at 0.

## Data Availability

Not applicable.

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
