# Peer review of "Cell Decision Making through the Lens of Bayesian Learning"

_entropy, 2023, doi:10.3390/e25040609_

Round 1

Reviewer 1 Report

Manuscript of Barua and Hatzikirou is devoted to the problem of cell decision as combination of uncertain cell inner state and external stimuli. Authors hypothesize that cell decision-making is implementation of Bayesian learning in diffusion-like (Fokker-Plank) process.

The text of ms has to be improved.

Lines in ms are enumerated with breaks (see, for example, page 4) that makes it more difficult to refer to. Critical points of the manuscript have to be explained better. Otherwise, I doubt that they are correct.

Major comments:

-          I do not understand why authors state that general power-laws and multimodal distributions are not normalizable? (page 3, explanation to eq. 1)

-          I do not understand derivation of Eq.2. a) The 3d line in equation has to be Hessian (second derivatives), but it is written squared first derivative. The approximation of KL over Fisher metric for shifted distribution is text-book one and can be directly referred to. b) The final expression is correct only in case of scalar parameter X.  c) If X is multidimensional, then Fisher metric is matrix and the result has to be quadratic form, which is wrong to write as a (deltaX)^2 * F. Therefore, transition from Eq.2 to Eq.3 is correct only in case of scalar state variable X.

-          The equation 5 is incomprehensible. Why X = t + beta * tau * (X – X’)? From where it comes from?

-          In steady state delta X ->0, therefore beta -> infinity. Therefore, in steady state we cannot learn any information about microenvironment from cell state, i.e. P(Y|X)=P(Y). I do not think that this is correct assumption.

-          Equation 8 and lines 92-93 are incomprehensible. What is exact form of entropy S? Is it equal ln(P(Y))?

-          In the paragraph, which starts after line 103, in eq.9, I got that M1 corresponds to X and M2 corresponds to Y. But in non-enumerated line after eq.11, authors state that M1 corresponds to Y and M2 corresponds to X.

-          Usage of Born–Oppenheimer approximation in eq.13 is questionable. The approximation was proposed in the model of direct physical influence of slow variable (nuclei motion) to fast variable (electron motion). Therefore, fixed in time slow variable constrains fast variable. In this case, slow variable (inner state of cell) cannot constrain evolution of fast variable (state of microenvironment).

-           

Minor point:

-          Reference before equation 5 is missed (line is not enumerated), as well as ref. on the end of page 5 (line is not enumerated as well).

-          Sentence “When the cell does not sense its microenvironment the internal state distribution is exponentially decaying” (line 150) has no sense. The whole paragraph lines 149-160 is incomprehensible.

Author Response

Reviewer 1:

Manuscript of Barua and Hatzikirou is devoted to the problem of cell decision as combination of uncertain cell inner state and external stimuli. Authors hypothesize that cell decision-making is implementation of Bayesian learning in diffusion-like (Fokker-Plank) process.

The text of ms has to be improved.

Lines in ms are enumerated with breaks (see, for example, page 4) that makes it more difficult to refer to. Critical points of the manuscript have to be explained better. Otherwise, I doubt that they are correct.

Our reply: We would like to thank reviewer 1 for his/her constructive comments, which had a great impact on the updated manuscript. Below, we address every comment and suggestion and incorporate it into the manuscript (with the red color) in a way we consider to be the most informative and understandable for the reader. In particular, we have heavily modified sections 1.1 and 1.2, along with adding Supplementary Information to detail some calculations. Regarding sec 1.2, we have decided to use a Gibbs ansatz solution for the P(X), assuming the independence of X components, which facilitates the analytical treatment of the problems. Also, we used these new derivations had implications for the whole paper and we had to adapt several other sections.

Major comments:

-I do not understand why authors state that general power-laws and multimodal distributions are not normalizable? (page 3, explanation to eq. 1)

Our reply: We would like to thank the reviewer for the comment since our formulation was not precise. Here we refer to distributions whose integral over the real space diverges, i.e. require finite support to be normalized. For instance, two power laws can have different support in order to be properly normalized. We have reformulated this point to “… distributions that require different finite support for their normalization

-I do not understand derivation of Eq.2. a) The 3d line in equation has to be Hessian (second derivatives), but it is written squared first derivative. The approximation of KL over Fisher metric for shifted distribution is text-book one and can be directly referred to. b) The final expression is correct only in case of scalar parameter X.c) If X is multidimensional, then Fisher metric is matrix and the result has to be quadratic form, which is wrong to write as a (deltaX)^2 * F. Therefore, transition from Eq.2 to Eq.3 is correct only in case of scalar state variable X.

Our reply: We would like to thank again the reviewer for drawing our attention to this valid point. The statement is correct and we have updated the Fisher information for the multidimensional case in the manuscript.

-The equation 5 is incomprehensible. Why X = t + beta * tau * (X – X’)? From where it comes from?

Our reply: There were some issues with this formula and we have removed this calculation as the other reviewers also mentioned. Indeed the multidimensional problem cannot be trivially solved.

-In steady state delta X ->0, therefore beta -> infinity. Therefore, in steady state we cannot learn any information about microenvironment from cell state, i.e. P(Y|X)=P(Y). I do not think that this is correct assumption.

Our reply: According to the comment we have redefined beta is proportional to the inverse value of delta X. The proportionality constant keeps beta finite.

-Equation 8 and lines 92-93 are incomprehensible. What is exact form of entropy S? Is it equal ln(P(Y))?

Our reply: We have heavily modified sec. 1.2 and clarified the steps required from the reviewer. We hope now it is clear.

-In the paragraph, which starts after line 103, in eq.9, I got that M1 corresponds to X and M2 corresponds to Y. But in non-enumerated line after eq.11, authors state that M1 corresponds to Y and M2 corresponds to X.

Our reply: It was a printing/typo mistake indeed. We have changed this in the draft.

-Usage of Born–Oppenheimer approximation in eq.13 is questionable. The approximation was proposed in the model of direct physical influence of slow variable (nuclei motion) to fast variable (electron motion). Therefore, fixed in time slow variable constrains fast variable. In this case, slow variable (inner state of cell) cannot constrain evolution of fast variable (state of microenvironment).

Our reply: Since this sentence is not essential for the derivation and is not completely clear to the reader, we have decided to remove it.

Minor point:

-Reference before equation 5 is missed (line is not enumerated), as well as ref. on the end of page 5 (line is not enumerated as well).

Our reply: This reference has been removed along with other parts of this section.

-          Sentence “When the cell does not sense its microenvironment the internal state distribution is exponentially decaying” (line 150) has no sense. The whole paragraph lines 149-160 is incomprehensible.

Our reply: This paragraph is essential in the discussion since intends to biologically translate our results. We have modified the text and we hope it is clearer now.

Reviewer 2 Report

The authors proposes a Bayesian learning model for cell decision making. 

I have found the theory to be logically consistent. Interesting finding between the LEUP theor yand Bayesian learning. 

I would suggest some minor check in mathematical expression. 

line 40 missing citation  please double check middle of page 3 equation dchi'dchi? equation 12 is there a typo on the first sigma line 104 missing citation  line 132 pdf maybe better spell out first for readers with less mathematical background line 152 extra e.g.   

Author Response

Reviewer 2:

The authors propose a Bayesian learning model for cell decision making. 

I have found the theory to be logically consistent. Interesting finding between the LEUP theory and Bayesian learning.

I would suggest some minor check in mathematical expression.

Our reply: We would like to thank reviewer 2 for his/her insightful comments. Below, we address every comment and suggestion and incorporate it into the manuscript (with the red color) in a way we consider to be the most informative and understandable for the reader.

line 40 missing citation please double check middle of page 3 equation dchi'dchi? Equation 12 is there a typo on the first sigma line 104 missing citation line 132 pdf maybe better spell out first for readers with less mathematical background line 152 extra e.g.

Our reply: We changed the draft according to the following comments. In line 152 we particularly tried to understand how cells adapt to their microenvironment via the reduction of conditional variance over time ( and reach a constant value ) such that they don't need further information about the microenvironment as they have learned/adapted to the microenvironment. In ref. [22], we showed previously in a setting of a flock ( in collective migration scenario ), each agent/cell updates its knowledge according to the microenvironmental entropy of the neighborhood cells (i.e., cells reside inside a particular sensing radius) which further tells us about the reduction of conditional variance over time.

Reviewer 3 Report

The authors present an interesting paper for a current field, namely decision-making. As I saw on the Internet, the authors have published on this topic. The article presents the applicability of the Bayesian learning hypothesis to which is added the derivation of the hierarchical Fokker-Planck equation for cell-microenvironment dynamics. The algorithm is interesting and quite well described.

To improve it, I recommend the authors:

- to complete the article with the Conclusions section from which the results obtained, why they differ from other similar results from the specialized literature and future research directions.

- in the Discussions section, the authors must make a comparison of the results obtained with other similar results from the specialty literature for decision-making applications, from which the contribution brought by the presented algorithm should be highlighted.

- if the authors have tested this algorithm on a real study, I recommend presenting the results, thus demonstrating its practical applicability, not just theoretical.

Author Response

Reviewer 3:

The authors present an interesting paper for a current field, namely decision-making. As I saw on the Internet, the authors have published on this topic. The article presents the applicability of the Bayesian learning hypothesis to which is added the derivation of the hierarchical Fokker-Planck equation for cell-microenvironment dynamics. The algorithm is interesting and quite well described.

Our reply: We would like to thank reviewer 3 for his/her insightful comments. Below, we address every comment and suggestion and incorporate it into the manuscript (with the red color) in a way we consider to be the most informative and understandable for the reader.

To improve it, I recommend the authors:

- to complete the article with the Conclusions section from which the results obtained, why they differ from other similar results from the specialized literature and future research directions.

Our reply: Our decision-making approach is a dynamic theory based on Bayesian learning of cellular internal states upon variations of the microenvironment distribution. The classical Bayesian decision-making methods are of static nature relying on Bayesian inference tools \cite{berger2013statistical}. Belief updating networks resemble the ideas of Bayesian learning, however, such algorithms are treated typically computationally, and to our knowledge, there have been not many attempts of deriving dynamic equations \cite{Jensen2007}. The oldest field where such ideas have been developed is human cognition. This date back to 1860 when Hermann Helmholtz postulated the Bayesian brain hypothesis, where the nervous system organizes sensory data into an internal model of the outside world \cite{Westheimer2008}. Recently, Karl Friston and collaborators formulated the brain free-energy theory deriving a variational bayesian framework for predicting cognitive dynamics. Friston's ideas have been recently translated into the Bayesian mechanics approach \cite{DaCosta2021}. The latter resembles to our approach however it requires concepts of Markov blankets and control theory. However, all the above refer to human cognition and not cell decision-making.

- in the Discussions section, the authors must make a comparison of the results obtained with other similar results from the specialty literature for decision-making applications, from which the contribution brought by the presented algorithm should be highlighted.

Our reply: Our answer above addresses both points.

- if the authors have tested this algorithm on a real study, I recommend presenting the results, thus demonstrating its practical applicability, not just theoretical.

Our reply: Unfortunately, the presented theory is rather theoretical at the moment. However, we are applying it currently to several biological cell decision-making systems such as T-cell differentiation, macrophage plasticity, and salmonella flagellation. Very soon we will have the first results of these biological problems.

References:

\cite{berger2013statistical}: Berger, J. O. Statistical decision theory and Bayesian analysis (Springer Science & Business 274 Media, 2013).

\cite{Jensen2007}: Bayesian Networks and Decision Graphs: February 8, 2007 109–166 (Springer New 346 York, New York, NY, 2007). doi:10.1007/978-0-387-68282-2_4. 347
\cite{Westheimer2008}: Westheimer, G. Was Helmholtz a Bayesian? Perception 37. PMID: 18605140, 642–650. 348 doi:10.1068/p5973 (2008). 349
\cite{DaCosta2021}: Da Costa, L., Friston, K., Heins, C. & Pavliotis, G. A. Bayesian mechanics for stationary 350 processes. Proceedings of the Royal Society A: Mathematical, Physical and Engineering 351 Sciences 477. doi:10.1098/rspa.2021.0518. arXiv: 2106.13830 (2021). 352

Round 2

Reviewer 3 Report

The authors responded to the requested clarifications.